# Determinants for Successful Digital Transformation

**Kyunghwan Oh [1], Hyeongseog Kho [2], Youngjin Choi [3] and Seogjun Lee [1],***

[1] School of Business, Konkuk University, Seoul 05029, Korea; barfool@koscom.co.kr
[2] Digital Marketing Team, Hyosung ITX, Seoul 07212, Korea; light211@gmail.com or richardokho@hyosung.com
[3] Department of Healthcare Management, Eulji University, Seongnam-si 13135, Korea; yuzin@eulji.ac.kr
* Correspondence: seogjun@konkuk.ac.kr; Tel.: +82-(24)-503645

**Abstract:** The proliferation of innovative digital technology is changing the industrial ecosystem; thus, companies should have the ability to adapt to the new environment. However, the success rate of digital transformation (DT) is still low, and there is a need to know its success determinant factors. This study aims to examine factors that affect DT's personal and social acceptance and empirically verify whether they actually affect it. Success factors and risk factors affecting the adoption of DT were identified from the literature review. The study collected data from 100 employees working for Korean financial institutions to statistically analyze and identify the determinant factors affecting successful DT. The results show that planned behavioral factors and innovative characteristics have a positive effect on DT acceptance attitude and that DT acceptance attitude has a positive effect on personal acceptance of DT. This study makes both theoretical and practical contributions. It distinguishes acceptance of innovation in two ways: individual acceptance and social acceptance, which has not been done in previous studies. It presents useful insights and understanding for those interested in transforming their organization with new technology by suggesting successful DT determinant factors.

**Keywords:** digital transformation; diffusion of innovations; acceptance attitude; personal acceptance; social acceptance

## 1. Introduction

The proliferation of innovative digital technologies is changing the industrial ecosystem. Innovative digital technologies such as the emergence of global platforms, new business models, and data-driven services through improved customer experience analysis are changing the way companies operate. Lau et al. [1] predicted that advances in digital technology will have direct or indirect economic effects in industries related to data production, consumption, and distribution, and that, by 2030, they will contribute to the global economic output by an additional $15.7 trillion. In addition to economic changes, industrial [2–4] and social [5,6] changes are also expected. As digital experience increases, consumers' requirements to reflect it continue to increase [7]. Consequently, the use of state-of-the-art technologies available in the market is essential for meeting consumer expectations and demands continuously [8]. Companies are hastily reorganizing their management strategies by setting digital-technology-driven service improvements as their primary goals, as consumer loyalty is linked to their survival [9].

Sallam et al. [4] predict that, due to accelerating digital transformation (DT), 75 percent of Fortune's top 500 global companies are expected to promote organizational changes through new technologies by 2025. Companies that fail to adapt to technological paradigm changes will disappear, and only those that evolve will survive [10]. Kreutzer [2] stresses that to survive in competition, companies must have the ability to adapt to a new environment. However, the success rate of DT is expected to be as low as 30% [11,12]. Companies understand the importance of DT, but implementation is still a challenge [13,14].

The strengths of digital technologies do not lie in individual technologies, and the integration and operation of digital technologies may vary depending on the digital maturity of the organization [15–18].

Behavioral studies in ICT-related [19,20] research areas have mostly focused on individuals' 'technology acceptance' rather than social change or social acceptance. On the other hand, previous studies in social research [21–23] areas have focused on the 'social change' theory, but little research has been conducted in terms of DT. The focus of this study is on both individual technology acceptance and social acceptance related to DT. Though previous studies [7,14,24–27] have suggested determinants for successful DT, few studies involved statistical verification of those determinants with empirical analysis. This study seeks to examine the determinant factors affecting personal and social acceptance of DT and to empirically verify those determinants using questionnaire data collected from Korean practitioners.

## 2. Literature Review

### 2.1. Definition of Digital Transformation

The Fourth Industrial Revolution began in Germany and the United States and influenced strategic government policies to lead the fourth industry around Europe [28–32]. The need for DT for the development and utilization of digital technology has been of much interest among enterprises, and DT adoption has begun to accelerate globally [2–4].

As various new digital technologies have emerged, the concept of IT-enabled transformation has been embraced in DT. Martin [33] defined DT as an individual-level technical ability to use digital technology and the perceived ability to know when to use it. White [34] defined DT as a business style change using new digital technology as a concept that emerges naturally due to changes in individuals, organizations, and society. He also explained that providing a digital work environment that integrates the four technologies of social, cloud, big data, and mobile (SCBM) can change the way work is done in terms of the productivity and competitiveness of individuals and organizations. Kane et al. [15] defined DT as a process of promoting technology adaptation by individuals, businesses, society, and countries, as well as the total social change caused by digitalization. Some researchers [24–26,35,36] defined DT from an organizational perspective, while others [27,37–39] defined DT from a social perspective.

According to Udovita [40], two distinct words are used for the expression of digitization: "digitization" and "digitalization". She said, "digitization" is simply a process of converting analog technologies, information, and products to digital formats, while "digitalization" creates new revenue and social or business value by changing business models or processes through digital opportunities. Udovita [40] defined DT as the reconstruction of changes in individuals and society that arise through digitization. Based on various prior studies on DT definitions, we defined DT as an "activity in which an organization makes social changes through customer-centered business model improvements using new digital technologies".

### 2.2. Theoretical Model

#### 2.2.1. Diffusion of Innovations Theory

DOI is a theory that explains how new ideas or technologies spread in society [41] and is a representative theory of acceptance and diffusion of innovations. Rogers [41] defined innovation as accepting ideas, practices, and objects that individuals or organizations recognize as new. Rogers explained that the degree of acceptance of innovation depends on how individuals recognize the characteristics of innovation. The acceptance of innovation involves a series of mental processes in which an individual or decision maker first recognizes innovation, forms an attitude, and decides to accept it; thus, diffusion of innovation is defined as a process that is passed on to members of society through a specific channel within a certain period [42].

DOI presents five characteristic variables for innovation: relative advantage, compatibility, complexity, trialability, and observability. Relative advantage refers to a criterion

for considering the profitability of innovation compared to existing innovations. Compatibility is a variable that indicates how much innovation corresponds to the values a person already has, and the lower the match, the more difficult it is for the innovative technology to be accepted. Complexity refers to the degree to which it is difficult to use innovative technology, and trialability refers to whether there is an opportunity to use it before accepting the actual innovation. Observability refers to how easy it is to see the use and consequences of innovation [41]. Moore and Benbasat [42], using the characteristics of the perceived innovation of DOI, set the variables as relative advantage, ease of use, image, visibility, compatibility, result demonstrability, and voluntariness and proposed the IDT, which measures the acceptance of individual innovation based on the validity of mutual verification. IDT is a model that supports the predictive validity of the DOI's perceived innovation characteristics [43], although it depends on how the individual recognizes the characteristics of innovation [44].

### 2.2.2. Extended Theory of Planned Behavior

Major theories related to innovation acceptance have been developed based on the theory of reasoned action (TRA), which describes human behavior. In the TRA, proposed by Fishbein and Ajzen [19], actual behavior is caused by behavioral intention, and the action intention is influenced by subject norms, meaning the perception of an individual's cognitive behavior and how people around them think about what they want to do. It also explains that attitudes toward behaviors are shaped by an assessment of the consequences of their actions, and subjective norms are influenced by the expectations of those around them and the motivation to comply with them [19].

Ajen [20] presented the theory of planned behavior (TPB) by adding a perceived behavioral control variable to TRA. Perceived behavioral control implies that the intention to act is influenced by resources and opportunities for action, such as time, money, skills, collaboration with others, and the ability to act [20]. Azen's [20] study explains changes in an actual behavior or behavioral intentions more accurately and contributes to completing the theoretical framework of human behavioral understanding and prediction, explaining the need for a modified approach to the extended theory of planned behavior (E-TPB) [20].

## 3. Research Design
### 3.1. Research Variable

This study aims to examine factors that affect the personal and social acceptance of DT and empirically verify whether they affect it. To this end, a literature review identified the determinant factors for successful DT. These factors were classified as behavioral factors at the level of personal behavior and innovative characteristics while comparing the characteristics of digital and existing technologies. In addition, the concept of acceptance of IDT was set as a dependent variable by dividing it into personal acceptance and social acceptance. The specific details of each study variable are as follows.

### 3.1.1. Behavioral Factors

The behavioral factors consist of four variables [41,45]: knowledge, individual innovativeness, self-efficacy, and involvement, derived from an investigation of individual action-level factors affecting DT.

Knowledge refers to accumulated experience related to a technology or product [46]. It is a variable that affects the relative advantage of the characteristics of innovation recognized in DOI. Rogers [41] stated that, the faster one figures out how to use new technology, the faster it is accommodated. Therefore, knowledge can be said to be the experience or level of understanding of digital technology. This knowledge is a basic factor in an individual's behavioral dimension as a first-phase variable in the innovation-decision-making process in DOI.

Individual innovativeness means that individuals are favorable to new technologies or products, like to use them, and tend to accept them before others [41]. It is a variable

that explains innovation at the individual level, one of the prerequisites of innovation in DOI. As individuals become more innovative, they are more open to products or services with new technologies, and those with higher individual innovation tend to embrace new technologies faster than others [47]. In the bell curve presented by Rogers [41], innovators and early adopters are highly innovative individuals and are also more likely to inform others around them about new technologies or products. In other words, an individual's innovation is a characteristic of the thinking that they are pioneers in some new technology or product [47].

Self-efficacy refers to the subjective judgment of an individual who believes that they could perform a task [43]. This is the same concept as the ease of use of IDT [42]. In other words, self-efficacy is the subjective judgment of an individual who is confident that digital technology can be used easily.

Involvement arises in conformity with perceived purposes based on the need, value, and interest in new technologies or situations, and is defined to include both emotional and cognitive relevance [48]. Involvement was first introduced in social judgment theory by Sherif and Hovland [21] as a variable developed in the field of social psychology. Involvement refers to the interest in a new skill or given situation or the value, relevance, and importance of a particular object [48]. Furthermore, it is common to conduct a study by dividing the level of interest or emotional attachment of the target into high and low, and the user's behavior may vary depending on the degree of involvement [49]. In the case of high involvement, attitudes are formed by a significant level of perceived effort around information through proven paths, but, in the case of low involvement, they are formed by peripheral information. High involvement acts to obtain fundamental information related to the product, and, in the case of low involvement, attitudes are formed by surrounding factors [50]. This study set a variable from the perspective of high involvement because the development of digital technology is of high personal and social interest.

### 3.1.2. Innovative Characteristics

Innovative characteristics consist of two variables: relative advantage and technological innovativeness.

Relative advantage is a criterion for thinking about how a beneficial innovation can be compared to a similar conventional method or technology [41]. It is a variable influenced by knowledge, a variable of the behavioral factors [51], and, the more people perceive the relative benefits of innovation, the greater the acceptance of innovation [41]. The perception that digital technologies will be more useful, convenient, reliable, and superior to traditional technologies is a relative advantage. The higher the relative advantage, the greater the acceptance of digital technologies. In other words, if digital technology provides value that is not found in existing technologies, the relative benefits increase.

Technological innovativeness interprets innovation at a technical level as a prerequisite for the innovation-decision-making process in DOI. It is a perception that the new technology is original and creatively different from existing technology [41]. Robertson [52] said that what was previously unseen was the process of newly recognizing and realizing thoughts, actions, and things. Lawton and Parasuraman [53] said that the newer the technology, the more innovative it is. Ram [54] said that novelty and innovation have different meanings, not all new technologies are innovative, and innovation must accompany change. In this regard, digital technology is not only new technology but also a great change, so it can be said that its technological innovativeness is high.

### 3.1.3. Personal Acceptance and Social Acceptance

Acceptance, a dependent variable of IDT, means accepting and using the value of a particular object. Attitudes change internally, and behaviors change externally. It is based on the premise that it is subject to acceptance, and, for a specific technology to be accepted, it needs an acceptable benefit and value even though there could be a loss. Acceptance can be classified into individual or social acceptance based on the scope of the impact. Social

acceptance means that the recipient is the general public [22,23]. The acceptance and use of the value of digital technology by members of the entire society can be called social acceptance. If it is within the scope of an individual, it can be called personal acceptance.

### 3.1.4. DT Acceptance Attitude

Fishbein and Ajzen [19] stated that actual actions are caused by intentions of action, which, in turn, are influenced by attitudes toward behaviors—cognitive factors of the individual. Davis [55] explained that the actual use of new technology arises from beliefs and intentions of use and that attitudes affect the acceptance of individuals or organizational members. In this study, acceptance attitudes were considered as variables mediating personal or social acceptance of innovation.

### 3.2. Hypothesis Development

Although our research model is based on E-TPB, the research model considers acceptance attitude as a mediating factor, not as an independent variable. Our research hypothesis assumes that behavioral factors and innovative characteristics affect acceptance behavior (both personal acceptance and social acceptance), and that effect is mediated by a person's acceptance attitude.

The following hypotheses were established based on previous studies. Figure 1 shows a visual depiction of our research model.

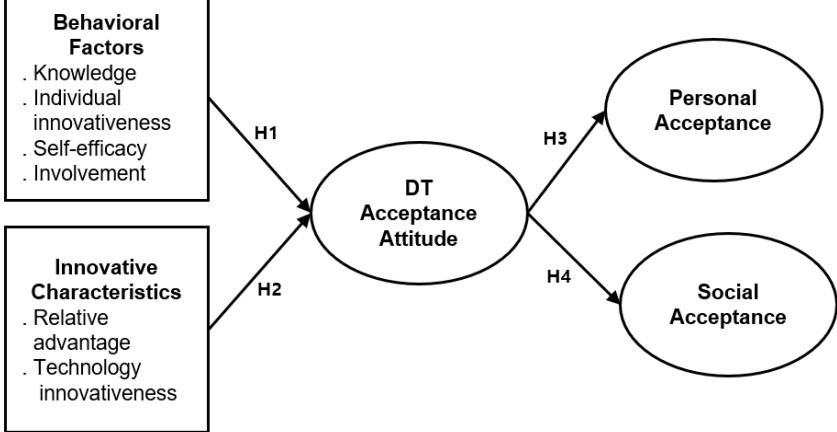

**Figure 1.** Research model.

The behavioral factor consisted of four variables: knowledge [41,46], individual innovativeness [41,56], self-efficacy [21,42,55–58], and involvement [48,49,55]. Knowledge is a starting variable in the first phase of the innovation-decision process in DOI and is a fundamental factor in an individual's behavioral dimension. Individual innovativeness is a variable that explains innovation at the individual level, a prerequisite for innovation in DOI. Self-efficacy is a concept similar to the ease of use of an IDT. Involvement is an extended concept of perceived value, a variable used in Zeithaml [59]. Based on previous studies, the following hypothesis is proposed.

**Hypothesis 1 (H1).** *The behavioral factors will have a positive (+) effect on DT acceptance attitudes.*

Innovative characteristics consist of two variables, relative advantage [41,47,50,60], and technological innovativeness [41,50,53], used in DOI and IDT. The following hypothesis is proposed based on prior research.

**Hypothesis 2 (H2).** *Innovative characteristics will have a positive (+) effect on the DT acceptance attitude.*

Based on a well-validated prior study with representative models of innovative acceptance, DOI, and IDT, this study seeks to verify how DT acceptance attitude affects both personal acceptance and social acceptance. Therefore, the following hypotheses are proposed:

**Hypothesis 3 (H3).** *DT acceptance attitude has a positive (+) impact on personal acceptance.*

**Hypothesis 4 (H4).** *DT acceptance attitude has a positive (+) impact on social acceptance.*

## 4. Research Methodology

### 4.1. Measurement Development

To investigate the factors affecting personal and social acceptance of DT, we set up the research variables. The questionnaire was designed to operationally define them. We constructed the measurement by making operational definitions of variables based on literature reviews, such as the measurements designed in the "Study on the advancement and determinants of the social acceptability model of intelligent information technology" published by KISDI in Son et al. [45] in 2019. The measurement consists of a total of 28 questions, excluding demographic questions, consisting of 11 for behavioral factors, 8 for innovative characteristics, 3 for DT acceptance attitude, 3 for personal acceptance, and 3 for social acceptance. All measurement items are ranked on a 7-point Likert scale (7 = very much, 1 = not at all). Table 1 shows the configuration of the measurement items that manipulate the definition of the variables.

**Table 1.** Operational definition of the variables.

| Variable | No. | Items | Related Studies |
|---|---|---|---|
| Construct 1: Behavioral Factors (BF) | | | |
| Knowledge | BF_1 | I am well aware of the pros and cons of products or services to which digital technology is applied. | Son et al. [45] |
| | BF_2 | I am well aware of products or services to which digital technology is applied. | Son et al. [45] |
| | BF_3 | I can explain to others about a product or service to which digital technology is applied. | Son et al. [45] |
| | BF_4 | I am confident in solving problems related to products or services to which digital technology is applied. | Moore and Benbasat [42] |
| Individual innovativeness | BF_5 | I usually use products with new technology before anyone else. | Son et al. [45] |
| | BF_6 | I try to use products or services with advanced technology first. | Son et al. [45] |
| | BF_7 | I tend to inform people around me about products with new technology. | Son et al. [45] |
| Self-efficacy | BF_8 | I think I can use digital technology more easily than others. | Son et al. [45] |
| | BF_9 | I think I can accumulate knowledge about digital technology in a relatively short time. | Son et al. [45] |
| | BF_10 | I am confident in using digital technology. | Son et al. [45] |
| Involvement | BF_11 | I am interested in innovative new digital technology. | Son et al. [45] |
| Construct 2: Innovative Characteristics (IC) | | | |
| Relative advantage | IC_1 | Digital technology is likely to be more useful than existing technology. | Son et al. [45] |
| | IC_2 | Using digital technology will be more convenient than using existing technology. | Moore and Benbasat [42] |
| | IC_3 | Digital technology is more reliable compared to existing technology. | Erjavec et al. [17] |
| | IC_4 | Digital technology will be better compared to existing technology. | Son et al. [45] |
| Technological innovativeness | IC_5 | I think digital technology is made with the latest technology. | Son et al. [45] |
| | IC_6 | Digital technology is innovative. | Son et al. [45] |
| | IC_7 | Digital technology is original, creative, and novel. | Son et al. [45] |
| | IC_8 | Digital technology differs greatly from existing technology. | Rogers [41] |

**Table 1.** *Cont.*

| Variable | No. | Items | Related Studies |
|---|---|---|---|
| **Construct 3: Digital Transformation Acceptance Attitude (DA)** | | | |
| DT acceptance attitude | DA_1 | I think positively about using products or services with digital technology applied. | Son et al. [45] |
| | DA_2 | I feel good about using products or services with digital technology. | Son et al. [45] |
| | DA_3 | I am actively in favor of the use of products or services to which digital technology is applied. | Son et al. [45] |
| **Construct 4: Personal Acceptance (PA)** | | | |
| Personal acceptance | PA_1 | I am willing to use a product or service with digital technology applied. | Son et al. [45] |
| | PA_2 | If I have a chance, I will use products or services with digital technology applied. | Son et al. [45] |
| | PA_3 | I will continue to use products or services with digital technology applied in the future. | Son et al. [45] |
| **Construct 5: Social Acceptance (SA)** | | | |
| Social acceptance | SA_1 | Digital technology and related products or services should be used more actively in our society. | Son et al. [45] |
| | SA_2 | Digital technology and related products or services should be used in more diverse areas of our society. | Son et al. [45] |
| | SA_3 | We need to gradually increase the use of digital technology in our society. | Son et al. [45] |

## 4.2. Participants and Data Collection

The survey in this study was conducted online from March to April 2021, targeting employees working for financial institutions in Korea. A preliminary survey of 125 employees was conducted to identify whether the respondents had a level of understanding of the exact concept of DT. We then distributed questionnaires to respondents. A total of 113 questionnaires were collected, and questionnaires with missing items were removed. Thus, 100 questionnaires were used for the analysis.

The results of the frequency analysis to examine the general characteristics of the survey respondents with basic data analysis are shown in Table 2.

**Table 2.** Respondent demographic characteristics (*N* = 100).

| Demographic Categories | | Frequency/Percentage |
|---|---|---|
| Age | Under 30 | 5 |
| | 30–39 | 75 |
| | 40–49 | 18 |
| | 50+ | 2 |
| Gender | Male | 66 |
| | Female | 34 |
| Financial institution classification | Bank | 59 |
| | Insurance company | 11 |
| | Financial investment company | 9 |
| | Financial assistance agency | 14 |
| | Other financial institutions | 7 |
| Position | Head of Department | 6 |
| | Senior Managers | 54 |
| | Junior Managers | 25 |
| | Others | 15 |
| Years of experience | 1–5 | 15 |
| | 5–10 | 48 |
| | 10–15 | 23 |
| | 15+ | 14 |
| Digitalization Progress level | Conceptual understanding | 13 |
| | Initial acceptance | 31 |
| | Initial diffusion | 35 |
| | Enterprise-wide diffusion | 17 |
| | Maturity | 4 |
| Total Responses | | 100 |

## 5. Data Analysis

### 5.1. Factor Analysis and Reliability

In this study, we used IBM SPSS Statistics Standard Version 27.0. Through the exploratory factor analysis (EFA) of variables, feasibility measurement items were grouped into factors, and the variables were simplified. The study ensures reliability using Cronbach's alpha coefficient. The feasibility and reliability analysis results of the research variables are the same as those in Tables 3–5. In addition, the Kaiser–Meyer–Olkin (KMO) values of all the variables were good, and, in the case of Bartlett's test of sphericity, the use of factor analysis was deemed appropriate.

**Table 3.** EFA results for the BF, IC construct.

| Construct | Item | Factor 1 | Factor 2 | Cronbach's Alpha |
|---|---|---|---|---|
| Behavioral Factors (BF) | BF_4 | 0.892 | 0.012 | 0.946 |
| | BF_8 | 0.865 | 0.103 | |
| | BF_10 | 0.851 | 0.114 | |
| | BF_3 | 0.831 | −0.011 | |
| | BF_2 | 0.828 | 0.078 | |
| | BF_5 | 0.821 | −0.077 | |
| | BF_7 | 0.818 | 0.015 | |
| | BF_9 | 0.795 | 0.115 | |
| | BF_6 | 0.783 | −0.003 | |
| | BF_1 | 0.717 | 0.139 | |
| | BF_11 | 0.612 | 0.396 | |
| Innovative Characteristics (IC) | IC_6 | −0.008 | 0.806 | 0.847 |
| | IC_4 | 0.035 | 0.793 | |
| | IC_1 | 0.219 | 0.767 | |
| | IC_7 | 0.070 | 0.739 | |
| | IC_2 | 0.110 | 0.739 | |
| | IC_3 | 0.258 | 0.690 | |
| | IC_8 | −0.023 | 0.661 | |
| | IC_5 | −0.137 | 0.616 | |
| Eigenvalue | | 7.273 | 4.478 | |
| % of variance | | 38.280 | 23.570 | - |
| Cumulative % | | 38.280 | 61.850 | |
| KMO and Bartlett's Test | | | | |
| Kaiser–Meyer–Olkin Measure of Sampling Adequacy | | | | 0.846 |
| Bartlett's Test of Sphericity | | Chi-Square | | 1556.737 |
| | | Degree of Freedom | | 171 |
| | | Significance | | 0.000 ** |

** $p < 0.01$.

**Table 4.** EFA results for the DA construct.

| Construct | Item | Factor 1 | Cronbach's Alpha |
|---|---|---|---|
| DT Acceptance Attitude (DA) | DA_2 | 0.874 | 0.918 |
| | DA_1 | 0.842 | |
| | DA_3 | 0.870 | |
| Eigenvalue | | 2.585 | |
| % of Var | | 86.178 | - |
| Cumulative % | | 86.178 | |
| KMO and Bartlett's Test | | | |
| Kaiser–Meyer–Olkin Measure of Sampling Adequacy | | | 0.759 |
| Bartlett's Test of Sphericity | | Chi-Square | 215.236 |
| | | Degree of Freedom | 3 |
| | | Significance | 0.000 ** |

** $p < 0.01$.

**Table 5.** EFA results for the PA, SA construct.

| Construct | Item | Factor 1 | Factor 2 | Cronbach's Alpha |
|---|---|---|---|---|
| Personal Acceptance (PA) | PA_2 | 0.904 | 0.376 | 0.954 |
| | PA_3 | 0.855 | 0.398 | |
| | PA_1 | 0.823 | 0.486 | |
| Social Acceptance (SA) | SA_1 | 0.320 | 0.897 | 0.945 |
| | SA_3 | 0.502 | 0.818 | |
| | SA_2 | 0.491 | 0.813 | |
| Eigenvalue | | 2.820 | 2.671 | - |
| % of Var | | 47.000 | 44.515 | |
| Cumulative % | | 47.000 | 91.514 | |
| KMO and Bartlett's Test | | | | |
| Kaiser–Meyer–Olkin Measure of Sampling Adequacy | | | | 0.844 |
| Bartlett's Test of Sphericity | | Chi-Square | | 753.226 |
| | | Degree of Freedom | | 15 |
| | | Significance | | 0.000 ** |

** $p < 0.01$.

### 5.2. Technical Statistical and Correlation Analysis

The normality verification of the measurement data is presented in Table 6. In addition, an analysis of the linear relationship between the variables using the Pearson correlation coefficient showed a clear difference between the variables in the study, such as in Table 7, the direction of the variables presented in the hypothesis, and a statistically significant correlation.

**Table 6.** Technical statistical analysis.

| Construct | N | Min | Max | Mean | SD | Skewness | Kurtosis |
|---|---|---|---|---|---|---|---|
| BF | 100 | 2 | 7 | 4.45 | 1.053 | 0.182 | −0.179 |
| IC | 100 | 3 | 7 | 5.39 | 0.901 | −0.108 | −0.553 |
| DA | 100 | 3 | 7 | 5.42 | 0.965 | 0.258 | −0.711 |
| PA | 100 | 3 | 7 | 5.66 | 1.116 | −0.079 | −1.283 |
| SA | 100 | 2 | 7 | 5.44 | 1.115 | −0.106 | −0.471 |

**Table 7.** Correlation analysis.

| | BF | IC | DA | PA | SA |
|---|---|---|---|---|---|
| BF | 1 | | | | |
| IC | 0.177 | 1 | | | |
| DA | 0.421 ** | 0.634 ** | 1 | | |
| PA | 0.373 ** | 0.594 ** | 0.839 ** | 1 | |
| SA | 0.291 ** | 0.623 ** | 0.788 ** | 0.801 ** | 1 |

** $p < 0.01$.

### 5.3. Hypothesis Test

The feasibility and reliability of the research variables were guaranteed through earlier analysis methods. Furthermore, factor analysis confirmed the reliability by conducting the subsequent simplified factor analysis. A multiple regression analysis was performed to verify this hypothesis.

### 5.3.1. Multiple Regression Analysis of Independent Variables and Intervening Variables

Multiple regression results on the effects of the independent variables of behavioral factors (BF) and innovative characteristics (IC) on the intervening variable of DT Acceptance Attitude (DA) are shown in Table 8.

**Table 8.** Multiple regression analysis of factors affecting DA.

| Variable | B | S.E | β | t | p | VIF | D-W | $R^2$ | Adj $R^2$ | F | Hypothesis | |
|---|---|---|---|---|---|---|---|---|---|---|---|---|
| Dependent Variable: DT Acceptance Attitude (DA) | | | | | | | | | | | | |
| Constant | 0.762 | 0.475 | - | 1.603 | 0.112 | - | | | | | - | |
| BF | 0.299 | 0.067 | 0.323 | 4.445 | 0.000 ** | 1.032 | 2.272 | 0.503 | 0.493 | 49.049 ** (0.000) | H1 | Adopted |
| IC | 0.618 | 0.078 | 0.577 | 7.936 | 0.000 ** | 1.032 | | | | | H2 | Adopted |

** $p < 0.01$.

First, the VIF (Variance Inflation Factor) index confirmation showed that there was no multicollinearity. Second, the Durbin–Watson index confirmed that all the variables were intact. Third, the value of model F is a statistically significant regression model, and this data is suitable for multiple regressions. Each of the above regression models shows that BF and IC all have a significantly positive (+) effect on DA, and hence, H1 and H3 are supported.

### 5.3.2. Multiple Regression Analysis of Intervening Variables and Dependent Variables

Next, the results of multiple regression analyses on the intervening variable DA and the dependent variables personal acceptance (PA) and social acceptance (SA) are shown in Table 9.

**Table 9.** Multiple regression analysis of factors affecting PA and SA.

| Variable | B | S.E | B | t | p | VIF | D-W | $R^2$ | adj $R^2$ | F | Hypothesis | |
|---|---|---|---|---|---|---|---|---|---|---|---|---|
| Dependent Variable: Personal Acceptance (PA) | | | | | | | | | | | | |
| Constant | 0.971 | 0.35 | - | 1.134 | 0.259 | - | 1.805 | 0.704 | 0.701 | 233.454 ** (0.000) | - | |
| DA | 0.917 | 0.064 | 0.839 | 15.279 | 0.000 ** | 1.000 | | | | | H3 | Adopted |
| Dependent Variable: Social Acceptance (SA) | | | | | | | | | | | | |
| Constant | 0.501 | 0.395 | - | 1.267 | 0.208 | - | 2.089 | 0.622 | 0.618 | 161.023 ** (0.000) | - | |
| DA | 0.911 | 0.072 | 0.788 | 12.689 | 0.000 ** | 1.000 | | | | | H4 | Adopted |

** $p < 0.01$.

After confirming that there were no abnormalities in the VIF index, Durbin–Watson, and model F values, multiple regression analysis showed that DA had a statistically significant impact on both PA and SA, and H3 and H4 were, therefore, adopted.

### 5.3.3. Parametric Regression Analysis

In addition to verifying our hypotheses, we also conducted a parametric regression analysis to identify whether DA has a mediating effect between independent and dependent variables using the three-step parametric regression analysis of Baron and Kenny [61]. The results of the analysis are shown in Table 10. Both the simple regression analysis of the first and second stages showed statistically significant results. Next, in the results of the three-step analysis, the independent variables were BF and DA, and the analysis when the dependent variable was PA showed that BF was not statistically significant, but DA was statistically significant. In addition, since the beta (β) value of the three-step independent variable, BF, is smaller than the second step, the mediating effect of DA is proved, and, in the third step, it has a complete mediating effect because it does not pay attention to PA, in which BF is a dependent variable. While DA had a complete mediating effect on PA, DA's mediating effect on SA was not verified.

**Table 10.** Three-step parametric regression analysis.

| | Independent Variable → Dependent Variable | | B | t | p | VIF | $R^2$ | F | Mediating Effect |
|---|---|---|---|---|---|---|---|---|---|
| Step 1 | BF | → DA | 0.421 | 4.598 | 0.000 ** | 1.000 | 0.177 | 21.142 ** | |
| | IC | → DA | 0.634 | 8.109 | 0.000 ** | 1.000 | 0.402 | 65.758 ** | |
| Step 2 | BF | → PA | 0.373 | 3.985 | 0.000 ** | 1.000 | 0.139 | 15.880 ** | - |
| | IC | → PA | 0.594 | 7.306 | 0.000 ** | 1.000 | 0.353 | 53.371 ** | |
| | BF | → SA | 0.291 | 3.008 | 0.003 ** | 1.000 | 0.085 | 9.051 ** | |
| | IC | → SA | 0.623 | 7.883 | 0.000 ** | 1.000 | 0.388 | 62.135 ** | |
| Step 3 | BF | → PA | 0.024 | 0.398 | 0.692 | 1.216 | 0.705 | 115.803 ** | Complete |
| | DA | | 0.829 | 13.630 | 0.000 ** | 1.216 | | | |
| | IC | → PA | 0.104 | 1.467 | 0.146 | 1.671 | 0.711 | 119.175 ** | Complete |
| | DA | | 0.774 | 10.960 | 0.000 ** | 1.671 | | | |
| | BF | → SA | −0.050 | −0.732 | 0.466 | 1.216 | 0.624 | 80.339 ** | Complete |
| | DA | | 0.810 | 11.790 | 0.000 ** | 1.216 | | | |
| | IC | → SA | 0.206 | 2.642 | 0.010 * | 1.671 | 0.647 | 88.914 ** | Partial |
| | DA | | 0.658 | 8.437 | 0.000 ** | 1.671 | | | |

\* $p < 0.05$, \*\* $p < 0.01$.

## 6. Discussion

Our study provided statistical proof that determinants suggested by previous research had significant effects on DA. The analysis showed that all the hypotheses were correct. H1, H2, and H3 represented consistent results with previous ICT-related research that both BF and IChad positive effects on DA, and in turn that the DA had a positive effect on PA. Furthermore, the mediating effect of DA was statistically significant. This finding is consistent with the TPB theory that users recognize and accept DT as a new technology.

Though H4 showed DA's positive effect on SA, the mediating effect of DA on SA was only partially accepted. While DA completely mediated the relationship between BF and SA, DA partially mediated the relationship between IC and SA. This finding implies that DA may be affected by other factors. So, further study is needed to better understand the relations among IC, DA, and SA.

Many studies on DT were conducted from the viewpoint that successful DT can be achieved along with individuals' successful 'new technology adoption'. However, our study tells us that organizations and society need 'social change' as well as new technology adoption to acquire successful DT. Thus, an integrated research model combining ICT-related research and social change research may be required to better understand the determinants for successful DT, because DT requires not only adopting new technologies but also diffusing those adopted technologies. Future research on the diffusion of innovation from a social perspective will need to involve the 'concept of resistance to change' and its effect on both PA and SA.

## 7. Conclusions

The purpose of this study was to investigate the determinant factors affecting personal and social acceptance of DT, and to empirically verify the effects of those determinants with questionnaire data collected from Korean practitioners. To this end, this study conducted a theoretical literature review including the Theory of Planned Behavior (TPB), Innovation Diffusion Theory (IDT), and Diffusion of Innovations Theory (DOI), etc. We then established research models and hypotheses to conduct factor analysis and regression analysis using the SPSS for empirical verification.

The study findings propose both theoretical and empirical implications. The study results support the "TPB (Theory of Planned Behavior)" that an individual's behavioral factors and innovative characteristics have a positive effect on the person's technology acceptance. Our empirical analysis showed that those effects were statistically significant, and that a person's acceptance attitude acted as a mediating factor between the independent and dependent variables.

Our empirical findings also showed that behavioral factors and innovative characteristics have positive effects on both individual and social technology acceptance. Regression analysis showed that DA completely mediated the effect on PA, while DA only partially mediated the effect on SA. These findings suggest that there can be additional mediating factor(s) that determine SA.

Our study contributes to both academic and practical perspectives. From an academic perspective, it adds more knowledge on how technology is socially spread by linking it to both individual acceptance and social diffusion. Previous researchers have considered individual acceptance and social acceptance separately and thus have not explicitly distinguished them. From a practical perspective, this study presents a useful insight for those in charge towards transforming their organization with new technology by suggesting the determinant factors for successful DT.

This study has some limitations. Although it considered both individual and social acceptance of DT, many of the determinant factors were derived from research conducted at the individual or organizational level. We presume that one can also learn a lot from studies conducted from a social perspective. To this end, more follow-up research should be conducted based on theories describing social perspectives. Follow-up research can also be conducted with questionnaires from various industries other than the financial industry that this study focuses on.

**Author Contributions:** Conceptualization, K.O., H.K., Y.C. and S.L.; methodology, K.O. and S.L.; software, K.O.; validation, K.O., Y.C. and S.L.; formal analysis, K.O. and S.L.; investigation, K.O.; resources, K.O., H.K., Y.C. and S.L.; data curation, K.O.; writing—original draft preparation, K.O.; writing—review and editing, K.O. and S.L.; visualization, K.O. and H.K.; supervision, S.L.; project administration, S.L.; funding acquisition, S.L. All authors have read and agreed to the published version of the manuscript.

**Funding:** This research received funding from Konkuk University.

**Institutional Review Board Statement:** This study is not applicable to any of the conditions that require Konkuk University's IRB review.

**Informed Consent Statement:** Informed consent was obtained from all subjects involved in the study.

**Data Availability Statement:** Not applicable.

**Acknowledgments:** This study was supported by Konkuk University in 2021.

**Conflicts of Interest:** The authors declare no conflict of interest.

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
