# Peer review of "Determinants for Successful Digital Transformation"

_sustainability, doi:10.3390/su14031215_

Round 1

Reviewer 1 Report

Simplify the title. Too long, not interesting.

Rewrite Abstract to include:

  • Background
  • Aim of the paper
  • Method adopted
  • Results
  • Contributions

Introduction

State explicitly your Research Questions. What are your curiosities here? Add a paragraph on what gaps you are addressing and thus the contributions to theory and practice your paper brings. In the last paragraph in the Introduction, please outline the structure of the paper.

The literature review is fine, but elaborate on why you are combining three big theoretical models DoI, MIR, and x-TPB? I know what they are and I am sure there is a compelling argument here and it’s not just for the sake of combining them.

So you need to argue the reasons in a paragraph. This is also to protect you from validation issues as your model has not been properly validated.

The research design is fine but redraw Fig 1, poor quality.

Methodology

Sample selection needs to be further elaborated. Why 100 is enough? How many companies? Are they really similar? How many were invited, how many were rejected. This needs to be comprehensive.

Data Analysis is fine.

Discussion.

What you have written here is actually part of the data analysis, as you talk about the data and the outcomes of your hypothesis testing. The Discussion section should be talking about the “big picture” and the “so what”.  Rewrite the Discussion section to reflect that. You can discuss the Implications for theory and implications for practice.

Conclusions

Please add limitations and future work here.

Ensure your manuscript is free from typos. Good luck.

Author Response

Simplify the title. Too long, not interesting.

  • Revised file Line 1
  • Changed the title: "Determinants for successful Digital Transformation"

Rewrite Abstract to include:

  • Background
  • Aim of the paper
  • Method adopted
  • Results
  • Contributions

  • Revised file Line 12~20
  • Rewrote Abstract according to your advice
  • "This study aims to examine factors that affect the personal and social acceptance of DT and to empirically verify whether they actually affect it. Success factors as well as risk factors affecting the adoption of digital transformation were identified from literature reviews. The study collected data from 100 employees working at Korean financial institutions to statistically analyze and identify determinant factors affecting successful DT. Study results show that planned behavioral factors and innovative characteristics have positive effect on DT acceptance attitude, and then the DT acceptance attitude have positive effect on personal acceptance of DT. The analysis also showed that innovative characteristics have negative effect on DT resistance, and then the DT resistance has negative effect on personal acceptance of DT."

Introduction

State explicitly your Re-search Questions. What are your curiosities here? Add a paragraph on what gaps you are addressing and thus the contributions to theory and practice your paper brings. In the last paragraph in the Introduction, please outline the structure of the paper.

  • Revised file Line 49-55
  • Rewrote and revise the introduction part according to your advice
  • "The purpose of this study is to examine determinant factors affecting the personal and social acceptance of DT and to empirically verify those determinants with questionnaire data collected from Korean practitioners. To this end, this study conducted a theoretical literature review including Innovation Diffusion Theory (IDT) and the Model of Innovation Resistance (MIR), and Diffusion of Innovations Theory (DOI) etc. We then established research models and hypotheses. to conduct factor analysis and regression analysis using the SPSS for empirical verification."

The literature review is fine, but elaborate on why you are combining three big theoretical models DoI, MIR, and x-TPB? I know what they are and I am sure there is a compelling argument here and it’s not just for the sake of combining them.

  • Revised file Line 56-149
  • Following your advice, literature review was abbreviated and reorganized into items related to our manuscript.  Stated our literature review focusing on "Extended theory of planned behavior", "Diffusion of Innovation", and Innovation Resistance"

The research design is fine but redraw Fig 1, poor quality.

  • Revised file Line 264
  • Redrew Fig 1 to present hypotheses and research variables

Methodology

Sample selection needs to be further elaborated. Why 100 is enough? How many companies? Are they really similar? How many were invited, how many were rejected. This needs to be comprehensive.

  • Revised file Line 337-343
  • A preliminary survey with 125 employees were conducted to identify whether the respondents have a level of understanding of the exact concept about DT.  We then distributed questionnaires to those respondents. A total of 113 questionnaires were collected, but questionnaires including missing items were removed and thus 100 questionnaires were used for the analysis.

  • Table 8-10
  • The VIF index confirmation showed that there was no multicollinearity. Furthermore, the Durbin-Watson index confirmed that all variables were intact. In addition, the value of model F is a statistically significant regression model, and this data was suitable for multiple regression.
  • There were no abnormalities in the VIF index, Dubin-Watson, and model F values

Data Analysis is fine.

Discussion.

What you have written here is actually part of the data analysis, as you talk about the data and the outcomes of your hypothesis testing. The Discussion section should be talking about the “big picture” and the “so what”.  Rewrite the Discussion section to reflect that. You can discuss the Implications for theory and implications for practice.

  • Line 433-469
  • Rewrote the Discussion as follows
  • This study distinguished acceptance of innovation into two ways; individual acceptance, and social acceptance. The study results show that in regards to individual acceptance both of the independent variables (E-TPS factors and Innovative characteristics) had positive effect on DT acceptance attitude & personal acceptance, and then the DT acceptance attitude also had positive effect on personal acceptance. Furthermore, innovative characteristics of a person had positive effect on resistance attitude, and then resistance attitude had positive effect on personal attitude.

  • In regards to social acceptance the analysis results show rather different implication. Personal attitude on both acceptance and resistance attitude did not have statistically significant effect on social acceptance. This finding implies that social acceptance for innovation needs difference types of determinant factors other than individually perceived characteristics. This implication gives us future research direction that follow-up research will need to be conducted focusing on theories describing social perspectives rather than theories describing individuals and organizational units.

    This study renders contribution in both academic and practical perspective. In academic perspective, this study suggests an understanding through which technology is socially spread both by linking individual acceptance and social diffusion, while previous researches have studied individual acceptance and social acceptance separately. In practical perspective, this study presents useful insight and understanding to those who try to innovate their organization with new technology by suggesting determinant lists for successful DT.

Conclusions

Please add limitations and future work here.

Ensure your manuscript is free from typos. Good luck.

Reviewer 2 Report

Dear authors,

Thank you for the opportunity to review the manuscript entitled "A Study on Factors Affecting Personal and Social Acceptance of Digital Transformation: Focusing on Attitudes and Resistance to Acceptance in the Financial Sector". This study explores the factors that affect the decision to adopt digital transformation at financial institutions. The authors adopted a online survey-based approach conducted from March 5-9, 2021. The authors find that the adoption of digital transformation increases if interest in digital technology is high, if perceptions of benefits of digital technology are more positive, and if there there is managerial, team-level, and organizational support to accommodate relevant changes. Although one can see the effort and work that has gone into your paper, I also do see several serious shortcomings of the paper in its current form:

  1. Clarity: This critique is connected to several aspects. First and foremost, you are trying to do much in one paper. You have fourteen hypotheses in one paper. You need to split the paper, and simplify what you are trying to achieve in this particular case. Second, related to the first point, you are trying to engage with too many streams of literature. You need to choose your audience, and engage exhaustively with your chosen theoretical perspective. Third, due to how much you are trying to achieve in one paper, readers have a hard time following your models for the different hypotheses and the related analytical procedures. Again, my recommendation would be to choose about four or five key hypotheses, and then focus on those.

  2. Contributions: The conclusions you draw from the study seem very generic and common-sense things applicable to almost every business setting. This seems to be exactly what a researcher and also a practitioner would expect and apply – thus, I am unsure about the contribution and novelty of your work.
  3. Language and grammar: Your manuscript needs language and grammar editing. The long blocks of dense text let readers lose sight of the key questions you are trying to address. This is more so because the transitions between your paragraphs are not smooth, rendering the reader unable to anticipate what is to come. You need to streamline your write-up by making it succinct, lucid, and easy to follow.

Author Response

Clarity: This critique is connected to several aspects.

First and foremost, you are trying to do much in one paper. You have fourteen hypotheses in one paper. You need to split the paper, and simplify what you are trying to achieve in this particular case.

Third, due to how much you are trying to achieve in one paper, readers have a hard time following your models for the different hypotheses and the related analytical procedures. Again, my recommendation would be to choose about four or five key hypotheses, and then focus on those.

[Line 83-316]

Simplified our research model;

From the previous model

To revised model (you can check the research model difference which is shown in Figures in the attached word file)

[Line 351-432]

Renewed our statistical analysis according to the revisied, simplified research model.

Second, related to the first point, you are trying to engage with too many streams of literature. You need to choose your audience, and engage exhaustively with your chosen theoretical perspective.

[Line 56-149]

Following your advice, literature review was abbreviated and reorganized into items related to our research focus.  Stated our literature review focusing on "Extended theory of planned behavior", "Diffusion of Innovation", and Innovation Resistance"

  • Contributions: The conclusions you draw from the study seem very generic and common-sense things applicable to almost every business setting. This seems to be exactly what a researcher and also a practitioner would expect and apply – thus, I am unsure about the contribution and novelty of your work.

[Line 433-469]

Rewrote the Discussion as follows

This study distinguished acceptance of innovation into two ways; individual acceptance, and social acceptance. The study results show that in regards to individual acceptance both of the independent variables (E-TPS factors and Innovative characteristics) had positive effect on DT acceptance attitude & personal acceptance, and then the DT acceptance attitude also had positive effect on personal acceptance. Furthermore, innovative characteristics of a person had positive effect on resistance attitude, and then resistance attitude had positive effect on personal attitude.

In regards to social acceptance the analysis results show rather different implication. Personal attitude on both acceptance and resistance attitude did not have statistically significant effect on social acceptance. This finding implies that social acceptance for innovation needs difference types of determinant factors other than individually perceived characteristics. This implication gives us future research direction that follow-up research will need to be conducted focusing on theories describing social perspectives rather than theories describing individuals and organizational units.

This study renders contribution in both academic and practical perspective. In academic perspective, this study suggests an understanding through which technology is socially spread both by linking individual acceptance and social diffusion, while previous researches have studied individual acceptance and social acceptance separately. In practical perspective, this study presents useful insight and understanding to those who try to innovate their organization with new technology by suggesting determinant lists for successful DT.

  • Language and grammar:Your manuscript needs language and grammar editing. The long blocks of dense text let readers lose sight of the key questions you are trying to address. This is more so because the transitions between your paragraphs are not smooth, rendering the reader unable to anticipate what is to come. You need to streamline your write-up by making it succinct, lucid, and easy to follow.

Thank you for your advice.

We’ve rewrote and revised the whole manuscript to make needed change.

I have marked the revised part of the language and grammar etc with “blue color” so that you can check which part was revised and how it was improved(?) according to your advice.

Reviewer 3 Report

A Study on Factors Affecting Personal and Social Acceptance of Digital Transformation: Focusing on Attitudes and Resistance to Acceptance in the Financial Sector.

Thank you very much for allowing me to review this paper. I think that the line of research between Digital Transformation and attitudes is very interesting.

After reviewing the paper, I believe there are some concerns that need to be resolved. More in detail:

1.     The introduction does not clarify what it is intended to investigate. In fact, the gap is not justified adequately. Why is it necessary to study this, and what it is intended to study? Please, could you clarify the gap and the connection with the literature?

2.     The literature review is well framed and described. However, I would like to see what your research model is (describing what relationships you intend to investigate). Additionally, I think you have to review more and current literature.

3.     Regarding the results, there is no discussion linking them with the literature review.

In sum the work requires some effort to clarify the gap, review the literature, research model, discussion of the results, and finally some more profound and new conclusions.

Author Response

Authors appreciate review's valuable comments. Reviewer's comment helped a lot in that all of the authors discuss several times to figure out what is the clear aim of our research, relevance of our research model, and the contents as a whole etc.

Revisions made according to reviewer's valuable advice are as follows:

1.   The introduction does not clarify what it is intended to investigate. In fact, the gap is not justified adequately. Why is it necessary to study this, and what it is intended to study? Please, could you clarify the gap and the connection with the literature?

In the 'introduction' part, [at page 2, line 55-64]. 

We've explained what this study is intended for, gap between the previous studies and research questions, focus of our study etc as follows:

Behavioral studies in ICT [46, 47] related research area have mostly focused on individual’s ‘technology acceptance’ rather than social change or social acceptance.  On the other hand, previous studies in social research [51,57,58] area have focused on ‘social change’ theory, but few research was conducted in terms of DT. The focus of this study is on both individual technology acceptance and social acceptance related to DT. Though previous studies [7,14,17,28,29,30,36] suggested determinants for successful DT, few studies involved statistical verification of those determinants with empirical analysis. This study seeks to examine the determinant factors affecting personal and social acceptance of DT and to empirically verify those determinants using questionnaire data collected from Korean practitioners.

2.     The literature review is well framed and described. However, I would like to see what your research model is (describing what relationships you intend to investigate). Additionally, I think you have to review more and current literature.

[At page 5, Line 231-234]

Explanation about our research model and what it is intended to study is added as follows:

Although our research model is based on E-TPB, the research model considers acceptance attitude as a mediating factor not as an independent variable.  Our research hypothesis assumes that behavioral factors and innovative characteristics affect acceptance behavior (both personal acceptance and social acceptance), and that effect is mediated by person’s acceptance attitude.

And according to your request to add more recent literatures, 9 more recent literature (literature 28-36) were added and explained as follows:

[at page 2, line 60]

Though previous studies [7,14,17,28,29,30,36] suggested determinants for successful DT, few studies involved statistical verification of those determinants with empirical analysis. This study seeks to examine the determinant factors affecting personal and social acceptance of DT and to empirically verify those determinants using questionnaire data collected from Korean practitioners.

[At page 2, Line 81-83]

Some researchers [28,29,30,31,32] defined DT from organizational perspective, while others [33,34,35,36] defined DT from social perspective.

[at page 14, line 498-519]

Recent literatures 28-36 are added in reference.

3. Regarding the results, there is no discussion linking them with the literature review.

[at 12, line 361-382]

Discussion was added in 5.3.4 to explain our research findings and implications, and the future research possibilities etc as follows:

Our study provided statistical proof that determinants suggested by previous researches had significant effect on DA. The analysis showed that all the hypotheses were accepted. H1, H2, and H3 represented consistent results with previous ICT related researches that both BF and IC had positive affect on DA, and in turn that the DA had positive affect on PA. Furthermore, the mediating effect of DA was statistically significant. This finding is consistent with the TPB theory that users recognize and accept DT as a new technology.

Though H4 showed DA’s positive effect on SA, the mediating effect of DA on SA was only partially accepted. While DA completely mediated the relationship between BF and SA, DA partially mediated the relationship between IC and SA. This finding implies that DA may be affected by other factors. So, further study is needed to better understand the relations among IC, DA and SA.

Many studies on DT were conducted from the viewpoint that successful DT can be achieved along with individual’s successful ‘new technology adoption’. However, our study tells us that organizations and society need ‘social change’ as well as new technology adoption to acquire successful DT. Thus, integrated research model combining ICT related research and social change research may be required to better understand determinants for successful DT, because DT requires not only adopting new technologies but also diffusing those adopted technology. Future research on the diffusion of innovation from social perspective will need to involve the ‘concept of resistance to change’ and its effect on both PA and SA.

In additions to reviewer's advice, the authors reviewed the whole manuscript and revised some contents that needs to be modified according to the reviewer's direction.  The revised contents are represented in "red color" so that the reviewer can identify the modified contents.

Informed consent information and questionnaire are also attached for your information. [at page 17, Line 585-670]

Round 2

Reviewer 1 Report

After the revision modification, this manuscript seriously suffers from a serious English problem. Please invest in paying a professional proofreader.

Take care of the use of articles 'a', 'an', and 'the'. Check the title.

2.1. Definitionon of Digital Transformation - poor spelling

"As various new digital technology has emerged" - should be various new digital technologies

And many other mistakes... please check this seriously.

Abstract: spell out DT as Digital Transformation the first time it is mentioned.

Rewrite Abstract to include:

  • Background
  • Aim of the paper
  • Method adopted
  • Results
  • Contributions

Add limitations and future work in the conclusion section.

Author Response

We appreciate your kind request.  Below are our revisions;

- Definitionon of Digital Transformation - poor spelling

"As various new digital technology has emerged" - should be various new digital technologies

And many other mistakes... please check this seriously.

Spell checks and other editing revisions are made as follows.

[Spell checks] in red collar

Line32 Transformation

Line67 Definition

Line 73 digital technologies have emerged

Line 428 Gartner

Line470 Industries

Line473 Government etc.

We proof-read the whole manuscript once more.  The revised parts are represented in blue color in the revised manuscript.

- Abstract: spell out DT as Digital Transformation the first time it is mentioned.

Spelled out DT as Digital Transformation the first time it is mentioned, and revised the abstract below.

Rewrite Abstract to include:

  • Background
  • Aim of the paper
  • Method adopted
  • Results
  • Contributions

Rewrote abstract as follows:

The proliferation of innovative digital technology is changing the industrial ecosystem, and thus the companies should have the ability to adapt to a new environment. The success rate of Digital Transformation (DT), however, is still low, and we need to know what are the determinant factors for successful DT. This study aims to examine factors that affect the personal and social acceptance of DT and to empirically verify whether they actually affect it. Success factors as well as risk factors affecting the adoption of digital transformation were identified from literature reviews. The study collected data from 100 employees working at Korean financial institutions to statistically analyze and identify determinant factors affecting successful DT. Study results show that planned behavioral factors and innovative characteristics have positive effect on DT acceptance attitude, and then the DT acceptance attitude have positive effect on personal acceptance of DT. This study renders contribution in both academic and practical perspective. In academic perspective, this study distinguished acceptance of innovation into two ways; individual acceptance, and social acceptance, while most previous researches have studied individual acceptance and social acceptance in each way separately. In practical perspective, this study presents useful insight and understanding to those who try to transform their organization with new technology by suggesting determinant factors for successful DT.

- Add limitations and future work in the conclusion section.

Added limitations and future work as follows:

There are limitations in this study. Although this study considered both individual and social acceptance of DT, many of the determinant factors were derived from researches conducted in individual or organizational level studies. We presume that one can also learn a lot from studies conducted in social perspective. To this end, follow-up research will need to be conducted based on theories describing social perspectives. Follow-up research can also be conducted with questionnaires from various industries other than financial industry that this study focused.

Reviewer 2 Report

Thank you very much for the opportunity to review your paper once again. It is clear that the authors exerted significant effort to improve the manuscript, particularly in terms of language and grammar, and improvement of the discussion section.  However, I do not believe that my review was addressed adequately. Specifically, the main issue was that the paper has too much going on, which affects clarity. The theoretical motivations behind setting up each of the twelve hypotheses remain unclear. To strengthen the paper, the authors need to clear why each of twelve hypotheses (and their alternative hypotheses) are relevant and central to the research question being addressed.

Author Response

We appreciate your kind request.  Below are our revisions;

Thank you very much for the opportunity to review your paper once again. It is clear that the authors exerted significant effort to improve the manuscript, particularly in terms of language and grammar, and improvement of the discussion section.  However, I do not believe that my review was addressed adequately. Specifically, the main issue was that the paper has too much going on, which affects clarity. The theoretical motivations behind setting up each of the twelve hypotheses remain unclear. To strengthen the paper, the authors need to clear why each of twelve hypotheses (and their alternative hypotheses) are relevant and central to the research question being addressed.

We appreciate your valuable advice and comments. As you requested, we have simplified and chose 4 key hypotheses, and then focuses on those 4 hypotheses. (mediating effect analysis were also conducted as statistical analysis, not as a hypothesis test)

As a result, we have revised all the way from abstract to conclusion, including simplified research model in Line 238~239, and then the following data analysis etc. The revised contents are in red color so that you can review and identify our modification.

We also proof-read the whole manuscript once more.  The revised parts are represented in blue color in the revised manuscript.

Reviewer 3 Report

The authors addressed to almost all my concerns.

However the conclusions need to be improved.

Which are the main theoretical and empirical conclusions?

Author Response

Authors sincerely appreciate reviewer's attentive and kind advice for revision.

Conclusion part was rewritten (in green color) including the theoretical and empirical contribution and conclusion of our study as follows:

The purpose of the study is to investigate the determinant factors affecting personal and social acceptance of DT, and to empirically verify the effect of those determinants with questionnaire data collected from Korean practitioners. To this end, this study conducted a theoretical literature review including Theory of Planned Behavior (TPB), Innovation Diffusion Theory (IDT) and Diffusion of Innovations Theory (DOI) etc. We then established research models and hypotheses to conduct factor analysis and regression analysis using the SPSS for empirical verification.

The study findings propose both theoretical and empirical implications. Study results support the “TPB (Theory of Planned Behavior)” that individual’s behavioral factors and innovative characteristics have positive effect on person’s technology acceptance. Our empirical analysis showed that those effects were statistically significant, and that person’s acceptance attitude acted as a mediating factor between independent and dependent variables.

Our empirical findings also showed that behavioral factors and innovative characteristics have positive effect on both individual and social technology acceptance. Regression analysis showed that DA completely mediated the effect on PA, while DA only partially mediated the effect on SA. These findings suggest that there can be additional mediating factor(s) that determine SA.

Our study contributes to both the academic and practical perspectives. From an academic perspective, it adds more knowledge on how technology is socially spread by linking it to both individual acceptance and social diffusion. Previous researches have considered individual acceptance and social acceptance separately and thus has not explicitly distinguished them. From a practical perspective, this study presents useful insight to those in charge of transforming their organization with new technology by suggesting the determinant factors for successful DT.

In addition, several minor manuscript revisions were made, which are shown in red color in the revised manuscript.